

# Brief communication: Evaluation of multiple empirical, density-dependent snow conductivity relationships at East Antarctica

Minghu Ding[1], Tong Zhang[1,2], Diyi Yang[1], Ian Allison[3], Tingfeng Dou[4], Cunde Xiao[2]

[1]State Key Laboratory of Severe Weather and Institute of Tibetan Plateau & Polar Meteorology, Chinese Academy of Meteorological Sciences, Beijing 100081, China
[2]State Key Laboratory of Earth Surface Processes and Resource Ecology, Beijing Normal University, Beijing 100875, China
[3]Antarctic Climate and Ecosystems Cooperative Research Centre, Hobart, Tasmania, Australia
[4]College of Resources and Environment, University of Chinese Academy of Sciences, Beijing 100049, China

*Correspondence to*: Minghu Ding (dingminghu@foxmail.com)

**Abstract.** Nine density-dependent empirical thermal conductivity relationships for firn were compared against data from three Automatic Weather Stations at climatically-different East Antarctica sites (Dome A, Eagle and LGB69). The empirical relationships were validated using a vertical, one-dimensional thermal diffusion model and a phase-change based firn diffusivity estimation method. The best relationships for these East Antarctica sites were identified by comparing the modeled and observed firn temperature at the depth of 1 m and 3m, and from the mean heat conductivities over two depth intervals (1-3m and 3-10m). Among the nine relationships, that proposed by Calonne et al. (2011) appears to have the best performance. This study provides useful reference for firn thermal conductivity parameterizations in land modeling or snow-air interaction studies on the Antarctica Ice Sheet.

## 1 Introduction

In the Earth's climate system, snow cover has two important physical properties, its high albedo and its low thermal conductivity. Both modulate heat exchange between the atmosphere and the surface (Dutra et al., 2010). Heat transportation in the near-surface snow layer plays a key role in controlling the upper thermal boundary condition of ice sheets (Ding et al., 2020).

Snow is a porous and inhomogeneous material with anisotropic thermal conductivity that is impacted by sphericity, dendricity, grain size, bond size, etc. ( Riche and Schneebeli, 2013). Direct measurements of snow heat conductivity can be made with a needle probe, heated plate and tomographic 3D images (e.g. Sturm et al., 1997; Calonne et al., 2011), all of which require intensive work. Alternative approaches include Fourier analysis methods that can estimate thermal diffusivity and reconstruct snow thermal histories from temperature measurements (Oldroyd et al., 20135), considering that the bulk/effective heat diffusivity can be more effectively described than the whole physical process of snow metamorphism. Similarly, the spatially averaged thermal diffusivity can be estimated from the changes of amplitude and phase of a temperature cycle with depth in the medium (Hurley and Wiltshire, 1993; Oldroyd et al., 2013). The numerical inverse



method (optimal control theory) is another possible approach for recovering thermal diffusivity by a least-squared method (Sergienko et al., 2008) or a recursive optimization approach (Oldroyd et al., 2013).

These numerical methods, however, need a relatively large number of temperature measurements, which can be difficult for large scale model studies. Thus, a widely-accepted alternative is to use laboratory-determined empirical
relationships to approximate the snow diffusivity and/or conductivity as a function of some typical and easily measured snow parameters such as snow density (e.g., Yen, 1981; Sturm et al., 1997; Calonne et al., 2011).

Density-dependent thermal conductivity relationships are widely used in various model studies. For example, the land model CLM and snow model SNTHERM use the empirical relationship developed by Jordan (1991), and is also adopted in other land surface energy balance and model studies, e.g., Wang et al. (2017). Lecomte et al. (2013) used the relationships in
Yen (1981) and Sturm (1997) for large scale sea ice-ocean coupling models. Applying the density dependent relationship in Calonne et al. (2011), Hills et al. (2018) investigated the heat transfer characteristics in the Greenland ablation zone. Steger et al. (2017) analyzed the melt water retention in the Greenland ice sheet by adopting the snow density-conductivity relationship given in Anderson (1976). Charalampidis (2016) used the relationship in Sturm (1997) to trace the retained meltwater in the accumulation area of the southern Greenland Ice Sheet. However, none of those relationships have been
carefully validated by in-situ data in Antarctica ice sheet.

In this paper, firn temperature data and snow density profile from three sites in East Antarctica were chosen to validate the applicability of these density/conductivity relationships (Table S1). We first describe the data collection at the three sites. After introducing the method for validating the empirical density-conductivity relationships, we then present the validation results, followed by discussions and conclusions.

## 2 Site and observational description

Several solar-powered automatic weather stations (AWS) have been deployed along Zhongshan to Dome A traverse route within the cooperative framework between Chinese and Australian Antarctic programs. These include deployments at LGB69 (in January 2002), EAGLE and Dome A (in January 2005). For more than 10 years since then, near-hourly meteorological measurements have been made of air and firn temperature (at several heights/depths), relative humidity, wind,
and air pressure. The data from the AWSs is remotely collected and relayed by the ARGOS satellite transmission system. Firn temperatures are measured (using FS23D thermistors in a ratiometric circuit with a resolution of 0.02K) at four depths below the surface. These were 0.1 m, 1 m, 3 m and 10 m when deployed, but they have slowly deepened with time due to snow accumulation. Due to heavy snowfall at LGB69, these data are only available for 2002-2008.

All three sites locate at western side of the Lambert Glacier Basin (Figure 1). LGB69 (70°50'S, 77°04'E; 1854 m a.s.l.)
is only 192 km away from coast (figure 1), and has an annual precipitation of 20 cm w.e. per year (~50 cm snowfall), strong wind (~8.5 m/s annually) and an annual air temperature of ~-26.10°C. EAGLE (76°25'S; 77°01'E) is a typical "surface glazed" area with hard snow crust because of the effect of drift snow. Its snow accumulation is 10 cm w.e. per year (30 cm snowfall) and the annual air temperature is ~-40.80 °C. Dome Argus (80°22′S, 70°22′E; 4093 m a.s.l.) is the highest point of





the east Antarctic Ice Sheet. It is also the summit of the ice divide of the Lambert Glacier drainage basin, ~1248 km from the nearest coast, and the surrounding region has a surface slope of only 0.01 % or less. Dome Argus has extremely low surface air temperature (-52.1°C annual average), humidity and snow accumulation rate (around 2 cm w.e. per year) and experiences no surface melt, even at the peak of summer (Ding et al., 2015). There were no shortwave and long wave radiation measurements at the site. Therefore, it is nearly impossible to build a complete energy balance model at the snow surface of Dome Argus.

## 3 Methods

1) *Numerical model method*. We validate the heat conductivity by a one-dimensional transient heat diffusion model,

$$\frac{\partial T}{\partial t} = \frac{K}{C_s \rho_s} \frac{\partial^2 T}{\partial z^2} \tag{1}$$

where $T$ is the firn temperature, $K$ is the heat conductivity, $C_s$ is the heat capacity of snow, $\rho_s$ is the density of snow, and $z$ is the depth below the snow surface. The vertical firn density profiles for three sites are shown in the Figures S4-S6. The heat capacity of snow is estimated by assuming snow is a mixture of air and ice,

$$C_s = C_i \frac{\rho_s}{\rho_i} + C_a \left(1 - \frac{\rho_s}{\rho_i}\right) \tag{2}$$

where $C_i$ and $C_a$ are heat capacity of ice and air, respectively. We constrain the upper and lower model domain by two Dirichlet boundary conditions, the 0.1 m and 10 m firn temperatures. The observed and modeled firn temperatures at the depths of 1 m and 3 m are then compared over a period of time. The performance of different heat conductivity relationships is then evaluated by the deviation metric of the difference between the modeled and observed temperature data,

$$\sigma^2 = \frac{1}{N} \Sigma_i^N (T_d - T_{dm}) \tag{3}$$

where $T_d = abs(T_{model} - T_{obs})$, $T_{dm}$ is the mean value of $T_d$, and N is the number of the temperature dataset.

2) *Temperature phase change method*. In this approach, we approximate the annual temperature cycles as sinusoidal functions (Demetrescu et al., 2007).

$$T(z, t) = T_m + A(z) \sin\big(\omega \cdot t + \varphi(z)\big) \tag{4}$$

where $T$ is the firn temperature expressed as a function of depth $z$ and time $t$, $T_m$ is the mean annual value of $T$, $A$ is the amplitude of the annual firn temperature cycle, $\omega$ is the frequency of the temperature cycle and $\phi$ is the wave phase of the annual cycle.

While it is common to fit a harmonic series (Fourier analysis) rather than a single sine wave to temperature variations, we found that this gave no advantage over Equation 4 for the data at the three sites because the temperature there have non-periodic temperature excursions during the "coreless" Antarctic winter. Assuming the snow is horizontally isotropic we can estimate the apparent thermal diffusion, $k_a$, from the changes of phase at different depths,

$$k_a = \frac{\omega}{2} \frac{(z_2 - z_1)^2}{(\phi_1 - \phi_2)^2} \tag{5}$$





The conductivity can then be recovered from $k_a$ and the heat capacity of firn.

**4 Results and discussions**

In Figures S1, S2 and S3, we show the comparisons of observed and modeled firn temperature using 9 different density-conductivity relationships at the Dome A, LGB69 and Eagle AWS station. The deviations of their differences are given in Table 1. At Dome A, the Ca2 relationship gives us a significant discrepancy between observed and modeled firn temperature

(Fig 2), probably because the density range corresponding to Ca2 is larger than that at Dome A (Fig S4), making Ca2 inappropriate for firn conductivity parameterization at Dome A.

In Figure 2 and Table 1, we can see that at Dome A, the Lan relationship gives the best performance at the depth of 1 m and 3 m, followed by the Van and Ca1 relationship. The Lan relationship was derived from in-situ snow conductivity measurements on Filchner Ice Shelf (Lange, 1985). It is the only relationship in this study that is based on in-situ Antarctica

firn sample measurements. The Ca1 relationshipwas derived by analyzing a wide range of different snow samples from a number of different geographical locations with many different snow types (Calonne et al., 2011). The Van relationship is old but is adopted in Cuffey and Paterson (2010), and still shows a nice performance in our model results.

Similarly to the Dome A case, at the Eagle station, the Ca1 relationship out-performs other relationships, followed by the Yen relationship. The Jor relationship, however, appears to have close performance to the Ca2 relationship, and is not

suitable for parameterizing the firn conductivity, compared with other relationships. A possible reason that the Ca2 relationship gives the most biased model results, compared to other density-empirical relationships, is because it is derived at a temperature level of around -3 °C and within a relatively high density range, whereas at the Dome A, LGB69 and Eagle Station, the snow density and temperature are both lower .

At LGB69, however, the Sch and Jor relationship appear to be superior to other relationships, different from the cases at

Dome A. The Jor relationship is based on the experimental measurements in Yen (1962). Sch is also an experimental relationship, based on the data given in Mellor (1977). In this case, the Ca2 relationship also gives the largest temperature difference. Note that the same relationship may have a different performance at different depth ranges. For example, for the Lan relationship, the modeled and observed firn temperature shows very good agreement at the depth of 3 m, but has a relatively large discrepancy at the depth of 10 m.

We also estimate the spatially averaged annual mean thermal conductivity from the temperature phase shifts between the depth ranges of 0.1 – 1 m, 1 – 3 m and 3 m – 10 m at the Dome A (Fig. S1), LGB 69 (Fig. S2) and Eagle (Fig. S3) Station, and compare them with the mean values corresponding to different density-conductivity relationships (Table 2). The phase at different levels ($\phi_{fit}$) (also the phase shift,$\Delta\phi_{fit}$) are determined from the least-squared fit to Equation (1), and the thermal diffusivity (conductivity) is then calculated by Equation (5).

Clearly, we can see in Table 2 that there is an increasing trend of conductivity with depth. Similar to the case in Table 1, the Ca2-derived conductivities show the largest discrepancy than the phase change-derived values, confirming again that the Ca2 relationship is not suitable for firn thermal simulations at these 3 stations in East Antarctica. In contrast, the Ca1





relationship gives much closer values than the conductivity values recovered from the phase change method at 3 different depth intervals, in consistent with the comparison results in Table 1. The Lan, Van and Yen relationships also show closer

agreement with the phase-change results, compared to other relationships (e.g., Ca2). However, we do not see a consistent pattern for the performance of different empirical density-conductivity relationships. At different depth levels, different relationships appear to have varying model performances. This is possibly a result of our assumption that the vertical density profile is kept constant in time, and that the heat capacity of firn is a linear relationship of the capacity of air and ice (Eqn 2). But since we consider a long time span of observation and model years (7 years for Dome A, 8 years for Eagle and 3 years

for LGB69), the overall deviation of modeled and observed temperature should be accountable in quantifying the performance of different density-dependent conductivity relationships.

## 5 Conclusions

In this study, we apply two methods to validate nine different density-conductivity relationships: 1) by applying a 1D vertical heat diffusivity model, we compare the modeled firn temperature at the depth of 1m and 3m with observations; 2)

we compare the mean empirical snow conductivity at three depth intervals (0.1-1m, 1-3m and 3-10m) according to the phase change derived temperature variations .

It is found that some empirical density relationships have generally good model performance and agree well with phase change recovered conductivity, but they show diverse behaviors at different depth levels. Based on the these two methods, we find that the relationship proposed by Calonne et al. (2011) (Ca1) generally has an overall best performance, while the

Calonne et al. (2019) relationship (Ca2) gives us the most biased comparison. The Jordan (1991) relationship (used in snow models like CLM and SNTHERM), however, does not present a very good model results for Dome A, LGB 69 and Eagle Stations. All in all, no conductivity-density relationship is optimal at all sites and the performance of each varies with depth.

The 3 AWS sites in different locations in East Antarctica that we have used for our validation cover a large range of elevation and distance from coast. We thus argue that our findings can shed some lights on firn thermal studies (e.g., the

applicability of different firn density-conductivity relationships) in Antarctica. We also urge for similar evaluations to be conducted at more geographic locations (e.g., west Antarctica Ice Sheet) where snow temperature and density observations are available.

## 6 Data availabity

AWS data are publicly available from http://aws.acecrc.org.au/ .

## 7 Supplement

The supplement related to this article is available online at: ******



## 8 Author Contributions

MD designed and wrote the paper, TZ did the calculation, DY and IA processed the AWS data, TD and CX evaluated the paper. All authors contributed to editing the manuscript.

## 9 Competing interests

The authors declare that they have no conflict of interest.

## 10 Acknowledgements

This study was funded by the National Key R&D Program of China (2019YFC1509100), the Strategic Priority Research Program of the Chinese Academy of Sciences (XDA20100300), the National Natural Science Foundation of China
(41771064) and the Basic Fund of the Chinese Academy of Meteorological Sciences (Grant Nos. 2018Z001 and 2019Z008). The observations in Antarctica were logistically supported by the Chinese National Antarctic Research Expedition (CHINARE).

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


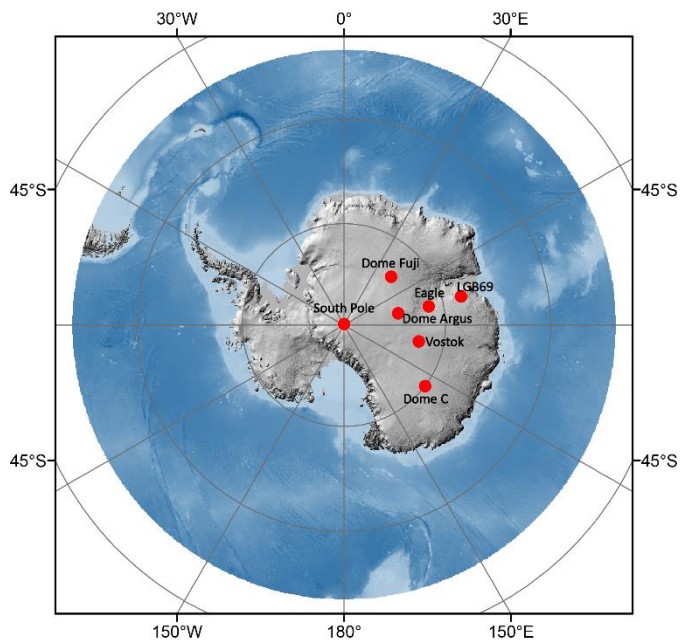

**Figure 1: The locations of Dome Argus, Eagle and LGB69 in Antarctica**




**Figure 2. Comparison of observed and modeled temperatures using different density-dependent conductivity relationships at the depths of 1 m (a, c, e) and 3 m (b, d, f) at Dome A (a, b), Eagle (c, d) and LGB69 (e, f).**






**Table 1. Deviation ($\sigma^2$) of $|T_{model} - T_{obs}|$ (K) for different density-dependent empirical relationships at 1 m and 3 m for three stations. The three overall best relationships for different depths are shown in blue.**

|       |     | Yen  | Ca1  | Jor  | Stu  | Lan  | Van  | Sch  | Ca2  | And  |
|-------|-----|------|------|------|------|------|------|------|------|------|
| Dome  | 1m  | 0.64 | 0.55 | 0.91 | 0.44 | 0.30 | 0.49 | 0.92 | 3.33 | 0.70 |
| A     | 3m  | 0.46 | 0.35 | 0.87 | 0.57 | 0.18 | 0.28 | 0.90 | 1.86 | 0.55 |
| LGB69 | 1m  | 0.33 | 0.36 | 0.22 | 0.56 | 0.04 | 0.44 | 0.19 | 0.49 | 0.30 |
|       | 3m  | 0.28 | 0.31 | 0.12 | 0.57 | 0.39 | 0.43 | 0.08 | 0.46 | 0.23 |
| Eagle | 1m  | 0.34 | 0.32 | 0.43 | 0.38 | 0.38 | 0.31 | 0.46 | 0.47 | 0.36 |
|       | 3m  | 0.14 | 0.12 | 0.33 | 0.38 | 0.69 | 0.19 | 0.39 | 0.33 | 0.14 |

**Table 2. Comparisons between density-dependent empirical conductivity and phase change recovered (PCR) conductivity (W m⁻¹ K⁻¹) at three depth intervals, 0.1-1 m, 1-3 m and 3-10 m. The three overall best relationships for different depths are shown in blue.**

|        |         | PCR  | Yen      | Ca1      | Jor      | Stu      | Lan      | Van      | Sch      | Ca2      | And      |
|--------|---------|------|----------|----------|----------|----------|----------|----------|----------|----------|----------|
| Dome A | 0.1-1m  | 0.26 | 0.30     | 0.28     | 0.38     | 0.18     | 0.22     | 0.26     | 0.39     | 0.04     | 0.32     |
|        |         |      | (+15.4%) | (+7.7%)  | (+46.2%) | (-30.8%) | (-15.4%) | (0%)     | (+50%)   | (-84.6%) | (+23.1%) |
|        | 1-3 m   | 0.31 | 0.33     | 0.31     | 0.42     | 0.20     | 0.29     | 0.28     | 0.43     | 0.10     | 0.35     |
|        |         |      | (+6.5%)  | (0%)     | (+35.5%) | (-35.5%) | (-6.5%)  | (-9.7%)  | (+38.7%) | (-67.7%) | (+12.9%) |
|        | 3-10 m  | 0.46 | 0.42     | 0.40     | 0.52     | 0.27     | 0.64     | 0.35     | 0.55     | 0.27     | 0.44     |
|        |         |      | (-8.7%)  | (-13.0%) | (+13.0%) | (-41.3%) | (+39.1%) | (-23.9%) | (+19.6%) | (-41.3%) | (-4.4%)  |
| LGB69  | 0.1-1m  | 1.02 | 0.47     | 0.44     | 0.58     | 0.32     | 0.91     | 0.39     | 0.62     | 0.36     | 0.49     |
|        |         |      | (-53.9%) | (-56.9%) | (-43.1%) | (-68.6%) | (-10.8%) | (-61.8%) | (-39.2%) | (-64.7%) | (-52.0%) |
|        | 1-3 m   | 0.62 | 0.50     | 0.48     | 0.61     | 0.35     | 1.16     | 0.41     | 0.66     | 0.42     | 0.53     |
|        |         |      | (-19.4%) | (-22.6%) | (-1.6%)  | (-43.6%) | (87.1%)  | (-33.9%) | (+6.5%)  | (-32.3%) | (-14.5%) |
|        | 3-10 m  | 0.81 | 0.59     | 0.57     | 0.72     | 0.45     | 2.34     | 0.49     | 0.79     | 0.57     | 0.63     |
|        |         |      | (-27.2%) | (-29.6%) | (-11.1%) | (-44.4%) | (+189%)  | (-39.5%) | (-2.5%)  | (-29.6%) | (-22.2%) |
| Eagle  | 0.1-1m  | 0.39 | 0.42     | 0.40     | 0.52     | 0.27     | 0.64     | 0.35     | 0.55     | 0.27     | 0.44     |
|        |         |      | (+7.7%)  | (+2.6%)  | (+33.3%) | (-30.8%) | (64.1%)  | (-10.3%) | (+41.0%) | (-30.8%) | (+12.8%) |
|        | 1-3 m   | 0.54 | 0.51     | 0.48     | 0.62     | 0.36     | 1.24     | 0.42     | 0.67     | 0.43     | 0.54     |
|        |         |      | (-5.6%)  | (-11.1%) | (+14.8%) | (-33.3%) | (+130%)  | (-22.2%) | (+24.1%) | (-20.4%) | (0%)     |
|        | 3-10 m  | 0.73 | 0.62     | 0.60     | 0.76     | 0.49     | 2.90     | 0.52     | 0.82     | 0.62     | 0.66     |
|        |         |      | (-15.1%) | (-17.8%) | (+4.1%)  | (-32.9%) | (+297%)  | (-28.8%) | (+12.3%) | (-15.1%) | (-9.6%)  |