# Peer review of "Brief communication: Evaluation of multiple empirical, densitydependent snow conductivity relationships at East Antarctica"

_The Cryosphere, 2021_

## Referee Comment (RC1)

This study presents a reliable evaluation on multiple empirical, density-dependent snow conductivity model/schemes, with three automatic weather station records. Although the subsurface heat flux is relatively low compared with the other components of air-snow/ice interaction, but it is essential for controlling the upper thermal boundary condition of ice sheets. As I know, there is urgent needs on the studies with in situ measurements in Antarctica. The sites the authors chosen can represents typical climatical regions of Antarctica, and they also presented a clear vision for further study, thus the result is effective and have a wide appeal. Several issues should be addressed prior to publication.

Line 15: "appears" should be "appeared".

Line 27: Oldroyd et al., 20135?

Line 27: histories or history?

Line 37-39: "For example, the land model CLM and snow model SNTHERM use the empirical relationship developed by Jordan (1991), and is also adopted in other land surface energy balance and model studies, e.g., Wang et al. (2017)." This words should be rewritten.

Line 46-49: The paragraph can be simplified and merged with the previous part.

Line 60: delete "figure 1".

Line 66: what is the lowest air temperature at Dome A? is it colder than Vostok?

Line 66: you may mean "specific humidity" rather "humidity"?

Line 67: "There were no radiation measurements at the site".

Line 75: I noticed that Figure S4-S6 showed before Figure S1 and suggested modification.

Line 103: "relationships".

Line 144-145: it is unexpected that Ca2 performed much worser than Ca1, what is the reason in your opinion?

Line 148-149: "The 3 AWS sites in different locations in East Antarctica that we have used for our validation cover a large range of elevation and distance from coast" can be "the 3 AWS sites in the paper cover a large range of elevation and distance from coast."

Line 150-152: "We also urge for similar evaluations to be conducted at more geographic locations (e.g., west Antarctica Ice Sheet) where snow temperature and density observations are available." Should be deleted.

Line168-216, Ensure the references format are consistent, such as line194,210, the publication years are different.

Line216, check the name of author, "Yen Y C" instead

Figure 2: the results of figure 2 is duplicated with Table 2, thus I suggest to move one of them into supplementary material.

Table 2: It is better to adjust the order of sites as "Dome A, Eagle and LGB69",

or "LGB69, Eagle and Dome A", which is similar with the figure 1. And this order is same with the introduction in "Results and discussions".

The grammar and writing in general is good enough for my understanding, but I am not a native English speaker, so I leave this issue to ED.

---

## Author Response (AR1)

Reply to RC 1

This study presents a reliable evaluation on multiple empirical, density-dependent snow conductivity model/schemes, with three automatic weather station records. Although the subsurface heat flux is relatively low compared with the other components of air-snow/ice interaction, but it is essential for controlling the upper thermal boundary condition of ice sheets. As I know, there is urgent needs on the studies with in situ measurements in Antarctica. The sites the authors chosen can represents typical climatical regions of Antarctica, and they also presented a clear vision for further study, thus the result is effective and have a wide appeal. Several issues should be addressed prior to publication.

Line 15: "appears" should be "appeared".

Authors: It has been modified.

Line 27: Oldroyd et al., 20135?

Authors: It has been modified.

Line 27: histories or history?

Authors: It is histories, as it is now.

Line 37-39: "For example, the land model CLM and snow model SNTHERM use the empirical relationship developed by Jordan (1991), and is also adopted in other land surface energy balance and model studies, e.g., Wang et al. (2017)." This words should be rewritten.

Authors: It has been modified into "For example, the empirical relationship developed by Jordan (1991) was adopted by the land model CLM, snow model SNTHERM, and many land surface energy balance studies, e.g., Wang et al. (2017)."

Line 46-49: The paragraph can be simplified and merged with the previous part.

Authors: A separate paragraph would better to illustrate the structure of the paper, we think.

Line 60: delete "figure 1".

Authors: It has been modified.

Line 66: what is the lowest air temperature at Dome A? is it colder than Vostok?

Authors: The lowest air temperature at Dome A was -82.3 ℃, which was recorded by an AWS at 10th July 2017. This is obviously higher than the lowest records of 89.2 at Vostok. However, the Landsat 8 has recorded a -93.2 value by remote sensing (https://www.nasa.gov/content/goddard/nasa-usgs-landsat-8-satellite-pinpoints-coldest-spots-on-earth). The Landsat 8 record still need a verification, I think.

Line 66: you may mean "specific humidity" rather "humidity"?

Authors: either "specific humidity" or "humidity" is ok here. It has been modified into specific humidity to avoid misleading.

Line 67: "There were no radiation measurements at the site".

Authors: It has been modified.

Line 75: I noticed that Figure S4-S6 showed before Figure S1 and suggested modification.

Authors: All figures and tables have been reordered.

Line 103: "relationships".

Authors: It is relationship for we only refer to Lan relationship.

Line 144-145: it is unexpected that Ca2 performed much worser than Ca1, what is the reason in your opinion?

Authors: As pointed by Reviewer 2, the Ca2 relationship is only suitable for deeply buried firn with densities from 550 to 917 kg/m3, which is not the case for the Dome A, LGB69 and Eagle station (density approximately ranges from 380 to 550 kg/m3). We now use another relationship for Ca2 as given by Calonne et al. (2019) and the new Ca2 relationship has a greatly improved performance.

Line 148-149: "The 3 AWS sites in different locations in East Antarctica that we have used for our validation cover a large range of elevation and distance from coast" can be "the 3 AWS sites in the paper cover a large range of elevation and distance from coast."

Authors: It has been modified.

Line 150-152: "We also urge for similar evaluations to be conducted at more geographic locations (e.g., west Antarctica Ice Sheet) where snow temperature and density observations are available." Should be deleted.

Authors: It has been deleted.

Line168-216, Ensure the references format are consistent, such as line194,210, the publication years are different.

Line216, check the name of author, "Yen Y C" instead

Authors: It has been modified. We also checked the reference through the context.

Figure 2: the results of figure 2 is duplicated with Table 2, thus I suggest to move one of them into supplementary material.

Authors: The Table 2 has been moved into supplementary.

Table 2: It is better to adjust the order of sites as "Dome A, Eagle and LGB69", or "LGB69, Eagle and Dome A", which is similar with the figure 1. And this order is same with the introduction in "Results and discussions".

Authors: It has been modified.

The grammar and writing in general is good enough for my understanding, but I am not a native English speaker, so I leave this issue to ED.
* * *
Reply to Reviewer 2

This study presents the evaluation of 9 density-based relationships of thermal conductivity applied to the upper part of the ice sheet column (0 -10 m). Relationships are evaluated based on data of in-situ measurements, including temperature profiles in firn, at three Antarctic sites. This work is of great interest for the firn community. The paper is clearly presented and reads well.

Authors: We thank the reviewers for the kind comments.

I have two comments prior publication.

1) It seems that the wrong relationship was taken from Calonne et al. 2019.   The formula Ca2 was used, that is only suited for deeply buried firn with densities from 550 to 917 kg/m3. Instead (or in addition), the evaluation of the relationship Equation (5) in Calonne et al. 2019 should be included, as it is a much-more suited formula to reach the goal of the present study. In Calonne et al. 2019, Equation (5) was suggested as a general formula to use for ice sheets, designed to work within the entire density range 0 – 917 kg/m3, from snow to firn to bubbly ice.   It actually combines relationship Ca1 and relationship Ca2, both evaluated in the paper. In addition, and in contrast to the others   evaluated relationships, Equation (5) allows to account for the temperature dependency on the thermal conductivity of firn through the choice of values for the thermal conductivities of pure ice ki and of pure air ka (for example, from Calonne et al. 2019: at -3°C, -20°C, and -60°C using ki= 2.107, 2.330, and 2.900 W/m/K and ka= 0.024,1460.023, and 0.019 W/m/K, respectively). The mean annual temperature

of firn within 0 – 10 m depth at the site could be used. This study would be a great opportunity to study the benefit of including the temperature dependency in the thermal conductivity modeling, with the three sites showing different mean annual temperatures.

Authors: We thank for the reviewer for pointing this out. We now adopt the relationship given by Eqn (5) in Calonne et al. (2019), and update Figure 2 and Tables 1, 2 (S2 of present version) and S1. The annual mean firn temperatures within 0-10 m at Dome A, Eagle and LGB site are used for calculating the heat conductivity of ice and air. Indeed, after using the new relationship, the updated Ca2 results are greatly improved and become much closer to observations and the Ca1 results. For example, at the Eagle Station, the Deviation ($\sigma^2$) of $|T_{model} - T_{obs}|$ decreases from 0.47 K to 0.32 K at the depth of 1 m, and from 0.33 K to 0.13 K at the depth of 3 m. The differences between Ca2 conductivity and phase change recovered (PCR) conductivity are also greatly decreased. Nonetheless, some old, density-dependent relationship like "Lan" and "Sch" appears to still out-perform the new temperature-dependent Ca2 relationship at some locations and depths, which may be attributed to a number of different reasons, for example, we use time-independent density profiles. Despite that, the new temperature-dependent Ca2 relationship gives one of the best model performances among all of these 9 relationships.

2) Thermal conductivity of firn depends largely on two parameters that are density and temperature. As the evaluated relationships were derived for different density and temperatures, their performances depend on which range of density and temperature is targeted. Thus, density range and temperature range in the area of interest (0 – 10 m depth) should be provided in the paper for the three sites. It is difficult to guess that only from the supplementary figures. The density range seems to range from 300 to 500 kg/m3 roughly. Comments in the discussion could be improved to explain the relationships' performance and link it to their domain of validity in density and temperature, when possible.

Authors: We thank the reviewer for underlying this point. Indeed, a lack of firn temperature in the density-dependent conductivity expression could be a large uncertainty in our evaluation results. We now add an introduction about snow density and temperature observations in the context between lines 60-70, and also revise and add some lines in the "Results and discussions" section, in order to clearly state the importance of firn temperature.

***** Minor comments

Line 20: "transportation" should be "transport"

Authors: It has been modified.

Line 23: Thermal conductivity of snow is anisotropic only for some snow types (depth hoar for example). Here I would suggest writing something like "Snow is a porous and inhomogeneous material with thermal conductivity that can be anisotropic and depends on the microstructure of snow: proportion of air and ice, grain shape, grain size, bonds size, etc."

Authors: The sentence is modified as suggested.

Line 27: "bulk/effective" should be "bulk/apparent".

Authors: It has been modified.

Line 27: " considering that the bulk/effective heat diffusivity can be more effectively described than the whole physical process of snow metamorphism" This sentence should be rewritten. Do you mean that this method assumes that all heat transport processes are represented thought the bulk thermal diffusivity, rather than accounting for all the different heat transport processes individually (heat conduction, heat convection, latent heat, for example)? If so, please note that some direct measurement method (e.g., the needle probe) also do this hypothesis. Maybe this part of the sentence should be simply deleted.

Authors: As suggested, we add a following sentence "as assumed also by needle probe measurement studies (Calonne et al., 2011)".

Line 37: "For example, the land model CLM, … ": this sentence should be rewritten.

Authors: The sentence has been rewritten.

Line 59 – 69: Figure S1 – S3 could be cited already in this paragraph, to illustrate temperature cycles at the three sites. Besides, that will allow citing Figure S1 to S5 in chronological order.

Authors: All figures and tables have been reordered.

Line 61: "8.5 m/s annually" should be 8.5 m/s mean annual"? Same for "annual air temperature" that should be "mean annual air temperature".

Authors: It has been modified and we also checked the other similar sentences.

Line 105: "relationship was".

Authors: It has been modified.

Table S1: first line of the table "-17 to -7" should be "-7 to -17".

Authors: It has been modified.

Line 110: "A possible reason that the Ca2 relationship gives the most biased model results, compared to other density-empirical relationships, is because it is derived at a temperature level of around -3 °C and within a relatively high density range, whereas at the Dome A, LGB69 and Eagle Station, the snow density and temperature are both lower." The Ca1 formula, that performs best, was also derived for a temperature of -3°C, so temperature cannot be the reason (this is relate to the above general comment on discussing relationship's performance regarding density and temperature range).

Authors: We now remove this sentence and make some clear statements that the temperature-dependent relationship Ca2 has better performance in some cases in the "Results and discussion" section.

Line 113: remove space before "."

Authors: It has been modified.

Line 141: remove space before "."

Authors: It has been modified.

Line 143: "Based on these two methods"

Authors: It has been modified.

---

## Editor Decision (ED1)

[revised manuscript text omitted]
 | 0.73 | 0.62 (-15.1%) | 0.60 (-17.8%) | 0.76 (+4.1%) | 0.49 (-32.9%) | 2.90 (+297%) | 0.52 (-28.8%) | 0.82 (+12.3%) | 0.62 (-15.1%) | 0.66 (-9.6%) |
| | | 31+19.234+9.7436.545+15.4499.35104258.85412.9783.7 | | | | | | | | | |